# Transcranial focused ultrasound to human rIFG improves response inhibition through modulation of the P300 onset latency

Justin M Fine*†, Archana S Mysore¶, Maria E Fini, William J Tyler‡, Marco Santello

School of Biological and Health Systems Engineering, Arizona State University, Tempe, United States

**\*For correspondence:**
justfineneuro@gmail.com

**Present address:** †Department of Neurosurgery, Baylor College of Medicine, Houston, United States; ‡Department of Biomedical Engineering, University of Alabama at Birmingham and UAB Heersink School of Medicine, Birmingham, United Kingdom

¶Co-first author

**Abstract** Response inhibition in humans is important to avoid undesirable behavioral action consequences. Neuroimaging and lesion studies point to a locus of inhibitory control in the right inferior frontal gyrus (rIFG). Electrophysiology studies have implicated a downstream event-related potential from rIFG, the fronto-central P300, as a putative neural marker of the success and timing of inhibition over behavioral responses. However, it remains to be established whether rIFG effectively drives inhibition and which aspect of P300 activity uniquely indexes inhibitory control—ERP timing or amplitude. Here, we dissect the connection between rIFG and P300 for inhibition by using transcranial-focused ultrasound (tFUS) to target rIFG of human subjects while they performed a Stop-Signal task. By applying tFUS simultaneously with different task events, we found behavioral inhibition was improved, but only when applied to rIFG simultaneously with a 'stop' signal. Improved inhibition through tFUS to rIFG was indexed by faster stopping times that aligned with significantly shorter N200/P300 onset latencies. In contrast, P300 amplitude was modulated during tFUS across all groups without a paired change in behavior. Using tFUS, we provide evidence for a causal connection between anatomy, behavior, and electrophysiology underlying response inhibition.

## Editor's evaluation

This study presents a valuable finding on the causal contribution of the inferior frontal gyrus (IFG) in behavioral control. State-of-the-art transcranial ultrasonic stimulation in combination with EEG is used to stimulate the IFG and find changes in speed and accuracy in a stop-signal task. This convincing work will be of interest to a wide range of basic neuroscientists.

## Introduction

Cognitive control allows a person to actively maintain and regulate goal-relevant thoughts and behaviors while suppressing context-irrelevant information (*Cohen, 2017*). The latter also entails overriding or stopping already initiated and prepotent actions and is often referred to as response inhibition (*Logan and Cowan, 1984*; *Wessel and Aron, 2017*). Response inhibition has been extensively studied given its centrality to human interactions with the environment and preventing adverse events (*Verbruggen and Logan, 2008*). Moreover, impaired inhibitory control has been associated with several neuropsychiatric disorders, for example attention-deficit/hyperactivity disorder (ADHD), impulse control disorders and addiction (*Bari and Robbins, 2013*). Hence, a deeper understanding of the neural dynamics and mechanisms of inhibitory control will contribute to developing better interventional techniques.

To identify neurophysiological markers of response inhibition, studies have utilized electro-encephalography (EEG) and functional magnetic resonance imaging (fMRI). EEG studies have provided substantial evidence about the fronto-central P300 event-related potential (ERP) as a key marker of response inhibition as it tracks the success of inhibitory outcomes (*Bekker et al., 2005*; *Greenhouse and Wessel, 2013*; *Kok et al., 2004*). Specifically, increased P300 amplitudes have been observed for trials that require response inhibition compared to trials that do not (*Enriquez-Geppert et al., 2010*). However, the P300 peaks after stop response timing — the stop signal reaction time (SSRT) — indicating that this amplitude modulation occurs too late to be an indicator of an inhibition response (*Huster et al., 2020*). Thus, an alternative interpretation is that the P300 amplitude may reflect outcome monitoring (*Huster et al., 2013*) or an attentional orienting process (*Corbetta et al., 2008*; *Polich, 2007*) that likely occurs after inhibition. Recent evidence has pointed to the P300 onset latency as being a candidate marker of response inhibition for two reasons: it tracks inhibition success and stopping speed (*Huster et al., 2020*; *Wessel and Aron, 2015*).

Although the P300 itself has a fronto-central topography in EEG studies, hinting at generation from medial prefrontal cortex (MPFC), areas comprising MPFC, such as pre-supplementary motor area, have been linked to inhibitory control and acting downstream from the right inferior frontal gyrus (rIFG; *Aron et al., 2016*). A plethora of fMRI and clinical lesions studies have identified the rIFG as a central node in triggering response inhibition (for review see *Aron, 2011*). This would suggest that rIFG might be responsible for the modulation of the P300 signal. However, this putative link between rIFG and P300 modulation remains to be determined. The reason for this gap stems from the issue that EEG and fMRI studies are mostly correlational in nature. These limitations have motivated others to use neuromodulation of brain areas in the inhibitory control network to establish their role (*Neubert et al., 2010*). For example, offline repetitive transcranial magnetic stimulation (rTMS) to rIFG during a stop signal task significantly disrupted inhibitory control by increasing SSRT (*Chambers et al., 2006*; *Chambers et al., 2007*). An EEG- rTMS study further reported that offline stimulation to rIFG reduced right frontal beta power and decreased inhibitory performance, thus establishing right frontal beta as a functional marker of inhibitory control (*Sundby et al., 2021*). These transcranial magnetic stimulation (TMS) studies further bolster rIFG as a central node in the inhibitory control network. However, the debate surrounding P300 timing as a reliable marker of inhibitory control and its connection to rIFG remains unresolved. Furthermore, when TMS is applied offline, it becomes challenging to identify the cognitive processing steps that are being perturbed in response inhibition.

The present study was designed to determine whether neuromodulation of rIFG can modify P300 amplitude, onset latency, or both while tracking behavioral inhibition. We simultaneously modulated rIFG activity with transcranial focused ultrasound (tFUS) during EEG recordings of subjects performing a Stop-Signal task (*Logan and Cowan, 1984*). Based on the above-reviewed evidence provided by EEG and TMS studies, we hypothesized that (1) P300 onset latency would track the speed of inhibitory control (SSRT) and (2) rIFG is causally related to inhibitory control through modulation of the P300 onset latency. Our findings corroborate these predictions, as we show that tFUS to rIFG improved stopping behavior by shortening the SSRT and P300 onset latency.

## Results

Human participants performed a Stop-Signal task while receiving time-locked, online tFUS on a subset of Go and Stop trials (*Figure 1*). Subjects were divided into groups according to tFUS stimulation type: (1) an experimental group that received active stimulation to right pars opercularis (rIFG), (2) a control group that received active stimulation to ipsilateral primary somatosensory cortex to account for non-site specific tFUS effects (S1), and (3) a second control group that received sham stimulation to account for possible tFUS-induced auditory artifacts (sham rIFG). tFUS simulation results are shown in *Figure 1—figure supplement 1*, with simulation outputs provided in detail in the methods. In the following, we use this stop signal task and tFUS to test the hypothesis that rIFG is causally related to inhibitory control through modulation of the P300 onset latency. According to this hypothesis, we predicted that the behavioral effects induced by tFUS should be limited to the alteration of stopping when tFUS is applied simultaneously with the Stop signal.

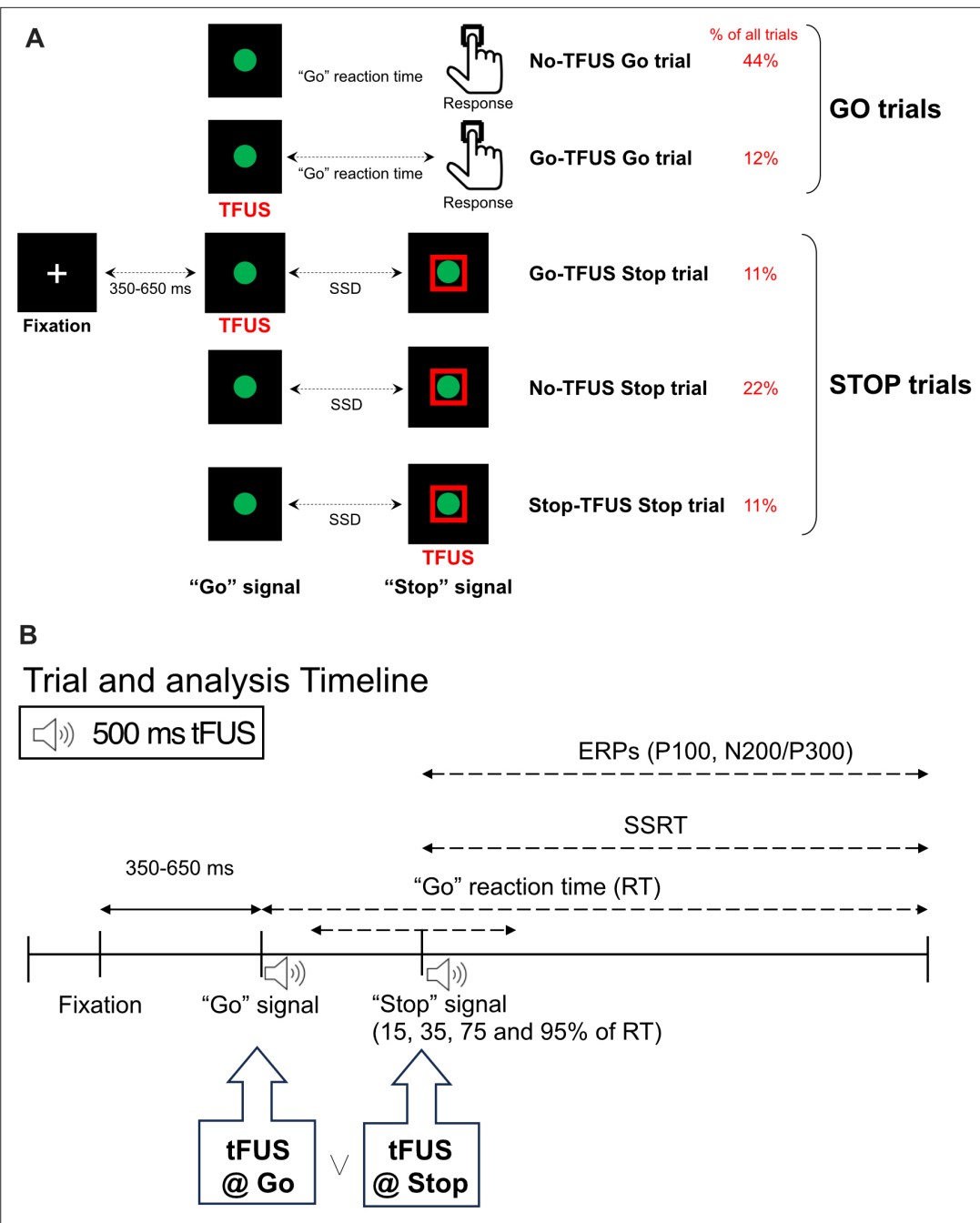

**Figure 1.** Task and trial design. (**A**) Stop-Signal task and trial types: rIFG group. Each trial type started with a fixation. After a randomly chosen delay (350–650ms), subjects were asked to respond to a "Go" signal as fast as possible by pressing the up key. On a subset of trials (Stop trials; rows 3–5), a red square appeared at one of 4 latencies (SSD) after the Go signal, cueing subjects to inhibit their response. Transcranial focused ultrasound (tFUS) was delivered to rIFG for 500ms, either at the onset of the Go (rows 2 and 4) or the Stop signal (row 5). SSD: stop signal delay. The same design was used for two control groups (S1 and sham rIFG), although tFUS was not delivered to a cortical site in the sham rIFG group (see text for details). For a subset of Go and Stop trials, no tFUS was delivered (rows 1 and 3). (**B**) The schematic further expands the display of the top plot, showing the timeline for each trial, with the timing of the events (Fixation, Go, and Stop signals). In addition, it shows how the core ERPs (P100, N200, and P300) are analyzed following the onset of the Stop signal. In trials that had tFUS, the tFUS was started simultaneously with the first monitor frame showing either the 'Go' or 'Stop' signal (aligned to t=0ms of the signal).

The online version of this article includes the following figure supplement(s) for figure 1:

**Figure supplement 1.** tFUS targeting and simulation.

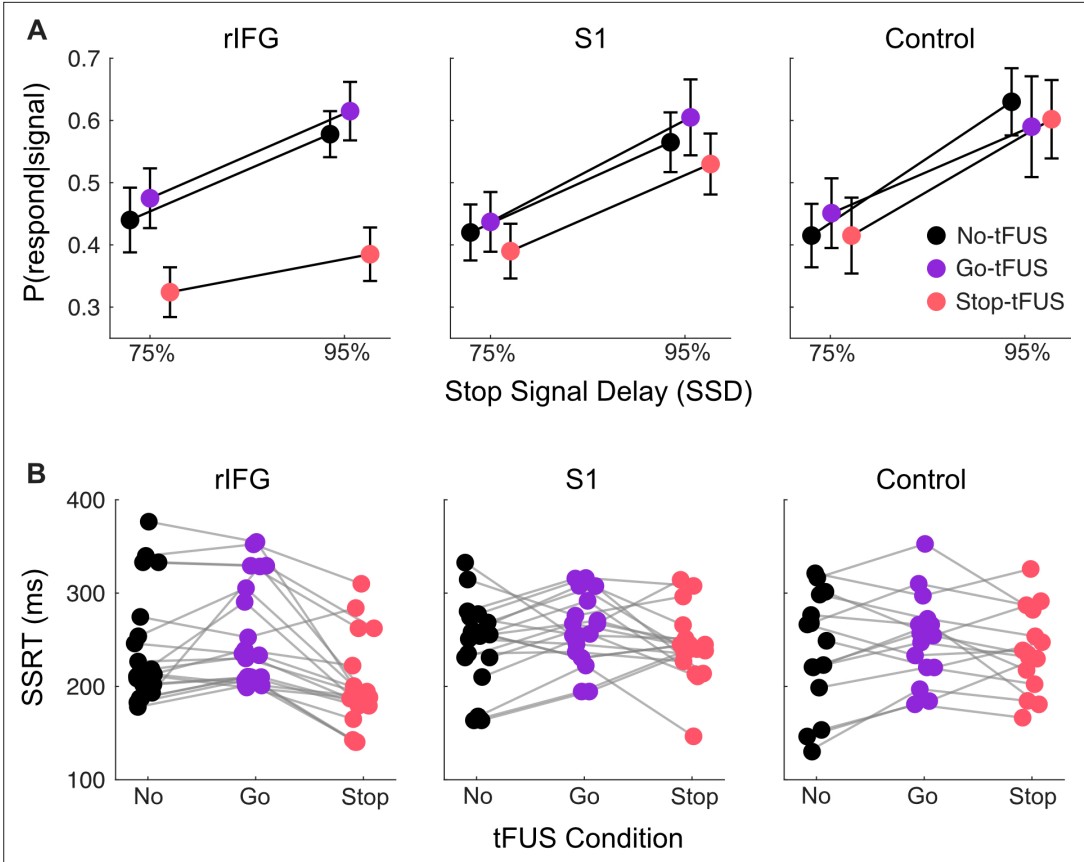

**Figure 2.** Response inhibition behavior. (**A**) Inhibition accuracy quantified as the probability of failing to inhibit conditioned on whether a stop-signal was presented, P(respond|signal). Results are shown for the 75% and 95% SSD conditions, for each tFUS condition (different colors), and for each group (columns, from left to right: rIFG group, S1 group, and control group). (**B**) Stop signal reaction times (SSRT) derived from using the Bayesian hierarchical fits for each subject (separate dots); each subject has three points, covering the three tFUS conditions. All groups are shown across the columns (from left to right: rIFG group, S1 group, and control group).

## tFUS to rIFG improves stopping behavior

We first addressed how the probability of failing to inhibit (P(respond|signal), *Figure 2A*) changed across tFUS conditions (No-, Go-, and Stop-) and groups. The main analysis of P(respond|signal) used a mixed-design ANOVA across the two highest SSD levels, 75% and 95% because these SSDs meet assumptions of the independent race model (see Materials and methods); we note that the same analysis including all SSDs did not show any tFUS-related changes in the shortest SSDs. Therefore, we present the rest of the results focusing on the 75% and 95% SSDs.

The ANOVA with the 75% and 95% SSDs indicated a significant interaction of tFUS and Group ($F_{(4,104)} = 7.83$, p<0.001, $\eta_p^2 = .23$), and main effects of tFUS ($F_{(2,104)} = 25.29$, p<0.001, $\eta_p^2 = .38$) and SSD ($F_{(1,52)} = 69.78$, p<0.001, $\eta_p^2 = .57$). First, the main effect of SSD is necessary, as we expect the P(respond|signal) should always go up with increasing SSD; this pattern can be seen across all groups in *Figure 3A*.

More importantly, the interaction of tFUS and Group indicates an impact of tFUS on inhibitory performance that was dependent on the Group. To source this interaction, we ran post-hoc t-tests to separately compare the No-tFUS condition to both Go-tFUS and Stop-tFUS P(respond|signal). The aim was to determine how tFUS in either condition (Go- or Stop-) deviated from non-stimulated stopping performance. The t-tests (Bonferroni corrected for number of tests) were run separately for each group and collapsed across SSD because there was no interaction with SSD and tFUS. Follow-up indicated a single difference of P(respond|signal) (*M*=0.15, SD = .10) within the rIFG group between No-tFUS and Stop-tFUS (*t*(20) = 6.84, CI$_{95\%}$ = [0.08, 0.23]). Overall, the main and follow-up analysis

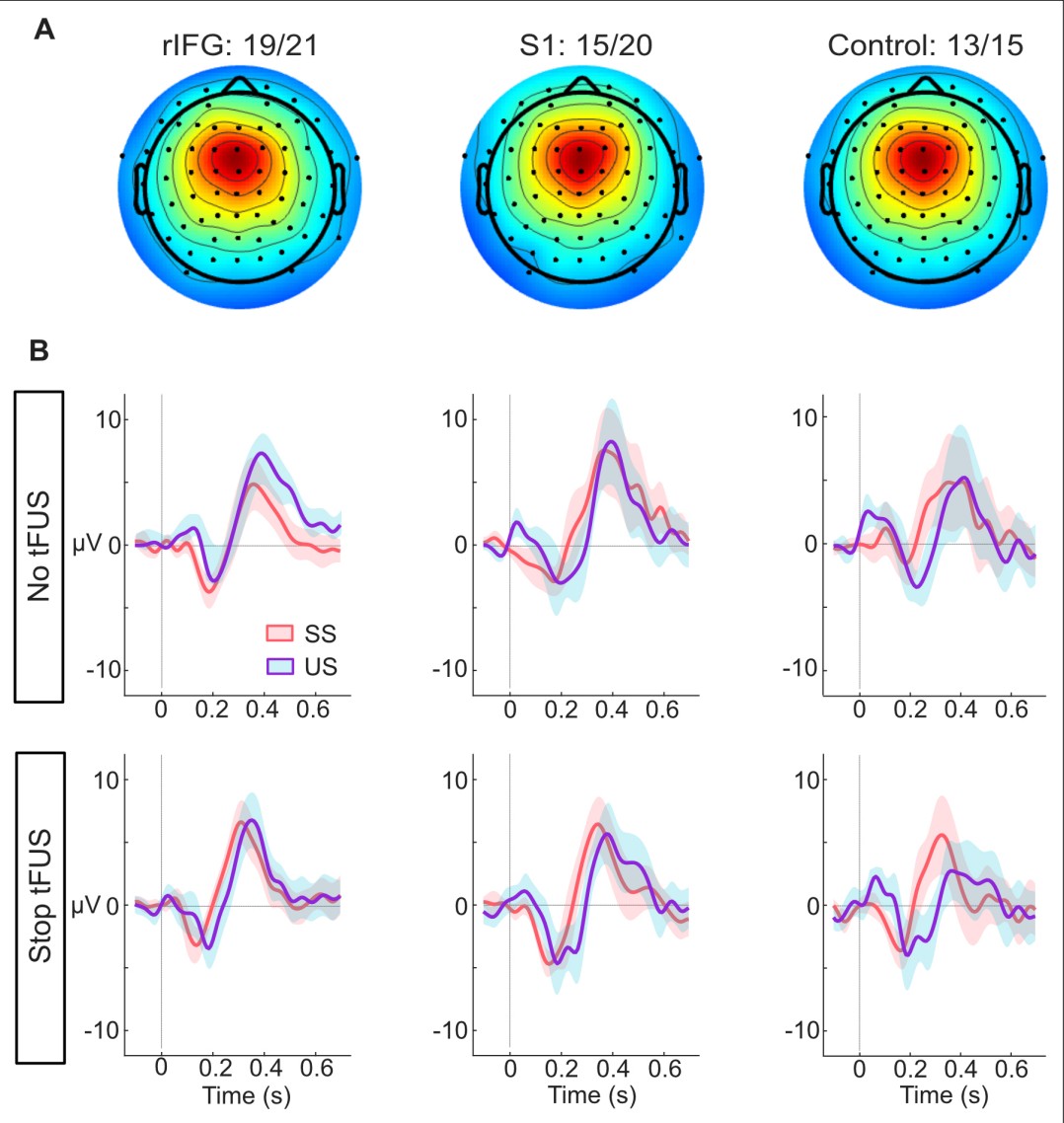

**Figure 3.** ERP responses modulated by tFUS. (**A**) Average medial frontal ICA spatial map per group, found from clustering. The number of subjects in each group that had a correspondent IC are listed above the maps.(**B**) Each plot in both rows (top and bottom) shows the within-group (per column) mean ICA-based ERP. The ERPs are plotted from –100ms before- to 650ms post-stop-signal onset, with the colors (orange and purple, respectively) corresponding to the successful (SS) and unsuccessful (US) stopping trials. The top row presents the rERPs for the No-tFUS conditions, and the bottom row are the rERPs for the Stop-tFUS condition. The vertical line denotes the time at which the Stop Signal was delivered.

indicate that tFUS had a targeted impact of improving stopping accuracy only in the rIFG group, and only when tFUS when applied simultaneously with the Stop signal. This is displayed clearly as a lower P(respond|signal) in Stop-tFUS compared to No-tFUS in *Figure 3A* (left column, showing rIFG group).

Next, we addressed the potential causes for the improvement in inhibitory performance. One possibility is that tFUS improved performance through a faster stopping process or faster stopping speed (SSRT). A mixed-design ANOVA with factors of tFUS (No, Go, Stop) and Group indicated a tFUS x Group interaction ($F(4,104) = 4.1$, $p=0.004$, $\eta_p^2 = 0.14$). Indeed, examining the mean SSRTs per Group and tFUS conditions (*Figure 3B*), we see that the SSRT during Stop-tFUS group is discernibly shorter than that in the No-tFUS condition for the rIFG group (*Figure 2B* left column). We corroborated this observation with two follow-up tests. First, using a simple main effects analysis over tFUS with Group as a moderator indicated the effect of tFUS on SSRT was only significant for the rIFG group ($F(2) =$

21.44, p<0.001). Second, we used t-tests (Bonferroni corrected for 6 tests) that compared No-tFUS SSRT individually to Stop-tFUS and Go-tFUS separately for each group. Only the rIFG No-tFUS versus Stop-tFUS ($M$=38ms, SD = 40ms; $t(20)$ = 4.40, $CI_{95\%}$ = [19ms, 68ms]) and Stop-tFUS versus Go-tFUS ($M$=61ms, SD = 42ms; $t(20)$ = 6.94, $CI_{95\%}$ = [32ms, 90ms]) tests were significant. Thus, this result points to tFUS improving inhibition through an impact on the speed of the stopping process.

Next, we addressed whether tFUS also affected the Go process independent of Stop trials, that is whether any effects found in P(respond|signal) could be explained by an altered, but independent, Go process. To do this, we used two mixed-design ANOVAs to separately analyze the mean and variability of GoO RTs (see Materials and methods). The two distributional moments for each subject were analyzed using a 3x2 mixed-design ANOVA with factors of Groups (3) and tFUS condition (2: No-tFUS, Go-tFUS trials). For both the Go RT mean and variability, we found no significant effect of tFUS Condition, Group, or their interaction (all p>0.05). These results suggest that neither tFUS (rIFG and S1 groups) nor auditory factors alone (sham rIFG group) altered Go RTs independent of a Stop signal. Put differently, any tFUS impact on inhibition (codified through P(respond|signal)) could not be explained by a modulation of the Go process.

Improvements in inhibition from tFUS could also have emerged from reduced mean or variability in the distribution of Go reaction times occurring during Stop Signal trials. Therefore, we analyzed SRRT, the Go response reaction times emitted during Stop trials. Two mixed-design ANOVAS were used to examine the subject-level means and variability with Group (3 levels) and tFUS (3 levels: No-tFUS, Go-tFUS, Stop-tFUS). We found no significant interactions or effects of tFUS on the mean (all p>0.05) or its variability (all p>0.05). Overall, these behavioral results indicate that only tFUS to rIFG improved response inhibition by shortening one or more processes related to the stopping speed.

## Neural responses underlying inhibition

Having found an acute impact of tFUS on behavior during Stop-tFUS trials in the rIFG group, next, we aimed to understand how these behavioral changes parlay into inhibition-related ERPs. We characterized neural activity by analyzing the mean activity in time windows around different medial frontal (*Figure 3A*) ERP peaks commonly found to modulate during response inhibition. These ERPs primarily include the N200 and P300 complex, as well as the P100. As seen in *Figure 3B*, we see all three of these ERPs across the groups. Considering our core hypotheses, in the mixed-design ANOVAs we used to analyze the ERP activity (see Materials and methods), our primary focus was the interaction of inhibition success and tFUS and their group differences. We next consider these ERPs in temporal order.

First, we start with the analysis of the P100 ERP. The ANOVA for the P100 indicated only a main effect of stopping success (F(1,46) = 37.88, p<0.001, $\eta_p^2$ = 0.29). A follow-up t-test comparing mean differences of the P100 between SS and US trials showed a larger mean amplitude during unsuccessful Stop trials ($M$=1.49 µV, SD = 1.71 µV; $t(45)$ = 5.92, $CI_{95\%}$ = [0.96 µV,1.94 µV]). This result suggests that, while P100 amplitudes were indicative of unsuccessful stopping, their modulation was not linked to performance changes induced by tFUS.

Next, we consider the results for the N200 ERP. We found a significant three-way interaction of Group, tFUS and stopping success (F(2,92)=4.91, p<0.01,$\eta_p^2$=0.11). We also found a main effect of stopping success (F(1,46) = 4.75, p<0.05, $\eta_p^2$ = 0.09). To interpret these results, we decomposed the interaction by using a simple effects analysis for each level of stopping success, with the tFUS and Group factors as moderators for post-hoc tests; all tests were Bonferroni corrected for the number of tests. This analysis showed that the three-way interaction was driven by a difference in N200 amplitude between tFUS (No and Stop) conditions successful stopping trials in the rIFG group. The negative amplitude was smaller during Stop-tFUS trials ($M$=–1.96 µV, SD = 1.82 µV; $t(19)$ = –3.88, $CI_{95\%}$ = [–3.71 µV,–0.32 µV]). This result indicates that Stop tFUS had a targeted effect of reducing N200 amplitude in the rIFG group.

Analysis of the P300 amplitude indicated only a marginal interaction of stopping success and tFUS (F(1,92)=4.05, p<0.05,$\eta_p^2$=0.06), but no Group effects. Visual examination of *Figure 4B* suggests that during No-tFUS trials, unsuccessful stopping had a larger peak, while Stop-tFUS trials points to similar peaks in successful and unsuccessful stopping trials. A follow-up simple main effects test comparing successful and unsuccessful trials with tFUS as a moderator, and collapsing across groups, confirmed this interaction. Unsuccessful stopping trials were characterized by a larger amplitude compared to

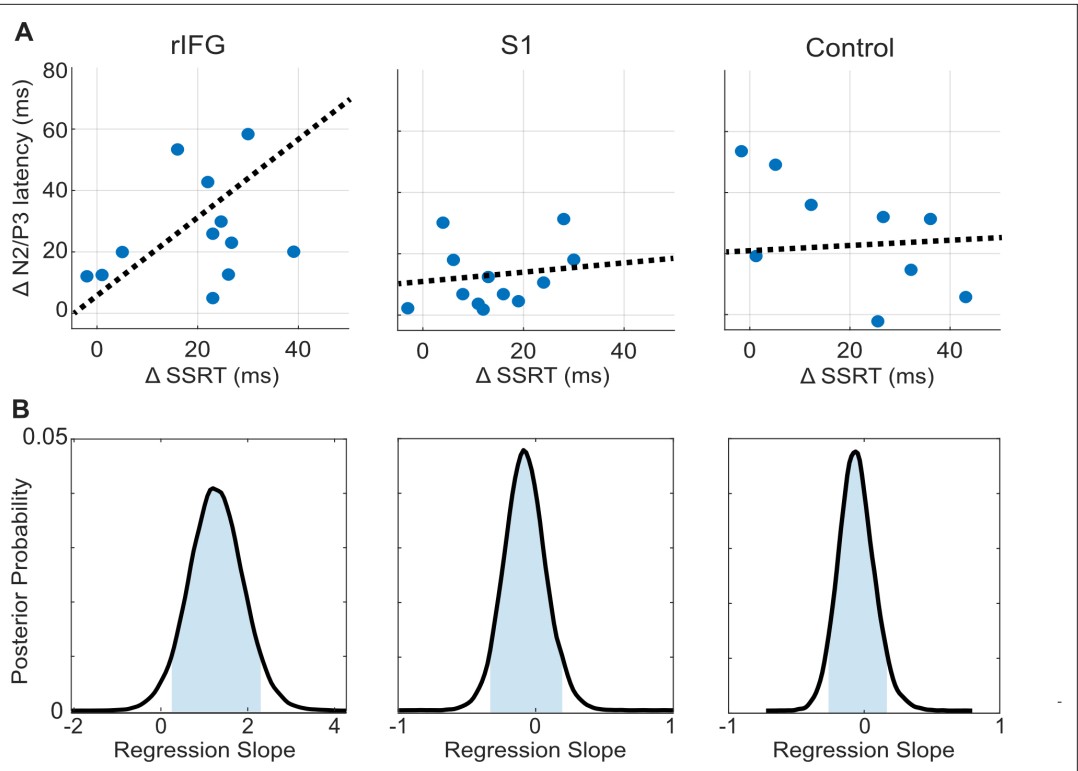

**Figure 4.** tFUS alterations to ERP timing modulate with SSRT. (**A**) Scatterplots of change in N200/P300 onset latency as a function of the change in SSRT between the No-tFUS and Stop-tFUS conditions. Dashed lines are the mean slope parameters from the posterior MCMC samples of each group's (separate columns) regression. (**B**). Each plot shows the kernel smoothed density (black line) of posterior MCMC samples for the regression coefficients and the 95% credible interval (blue shaded area).

successful trials, but only in the No-tFUS conditions (F(1,45)=4.21, p<0.05); however, this effect was not significant after adjusting for both tests. These results suggest any significant changes in P300 amplitude were relatively weak and insensitive to the changes in inhibition performance found in the rIFG group.

Finally, we examine the N200/P300 onset latency correlation with SSRT as a means to identify a clearer connection between rIFG group performance changes during Stop-tFUS. Specifically, we tested the prediction that the change in SSRT accompanying Stop-aligned tFUS to rIFG would co-modulate with the onset latency of the N200/P300 ERP complex. To do this, we regressed the difference in SSRT between No-tFUS and Stop-tFUS conditions against the difference in onset latency between the same conditions. We used a Bayesian approach to estimate the posterior probability of a non-zero coefficient (see Materials and methods and *Figure 4B* for posterior samples) for each group. In line with the behavioral change in SSRT in rIFG, we found that the change (regression slope) in SSRT co-modulated with ERP onset latency only in the rIFG group (*Figure 4A*, column one). Only this group showed a posterior mean slope having a 95% credible interval outside of zero (*Figure 4B*). Both the S1

**Table 1.** The mean and standard deviation (SD), 95% credible interval tails of the regression parameters of change in SSRT against change in N200/P300 onset latency, taken from the posterior sampling distribution for each group.

| Group | Mean | SD | 5% Credible interval | 95% Credible interval |
|---|---|---|---|---|
| rIFG | 1.27 | 0.43 | 0.23 | 2.32 |
| S1 | 0.13 | 0.24 | –0.12 | 0.31 |
| Control | 0.15 | 0.21 | –0.17 | 0.48 |

and control group had slope parameters that overlapped with zero (*Figure 4B*, *Table 1*). This analysis provides a more direct link between rIFG tFUS and stopping control through speed.

## Discussion

The present study builds on a rich literature that has considered response inhibition from many perspectives. By employing online tFUS to rIFG in parallel with EEG, we isolated N200/P300 onset latencies as the primary predictor of behavioral outcomes – response inhibition performance via SSRT – during No-tFUS and Stop-tFUS trials. This result is consistent with recent work suggesting P300 onset latencies predict the SSRTs (*Huster et al., 2020*; *Huster et al., 2013*; *Wessel and Aron, 2015*), with latency modulation occurring before the SSRT. This close temporal proximity of neural modulation and SSRT is predicted by Stop-Signal task studies of single units in non-human primates (*Hanes et al., 1998*), as well as attractor network (*Lo et al., 2009*) and related accumulator models (*Boucher et al., 2007*; *Logan et al., 2015*). By directly stimulating and measuring rIFG simultaneously, while showing a concurrent change in P300 onset and behavior in rIFG but not control groups, we established a causal connection between inhibitory control, P300 timing, and rIFG.

In terms of neurophysiology, several studies have presented contrasting results on whether the P300 or the N200/P300 onset latency is a valid marker of response inhibition. Several of our results favor a framework in which the P300 onset latency tracks the inhibition process. For example, we found the P100, N200, and the P300 all differentiated successful versus failed inhibition. However, the P100 differences were insensitive to tFUS-induced changes in rIFG group inhibition. The N200 interaction of successful stopping and tFUS specific to the rIFG group would suggest the relevance of this ERP amplitude to inhibition. However, under the assumption that the N200/P300 is a complex of interacting signals, we suspect the reduced N200 amplitude during SS trials during Stop-tFUS in the rIFG group is the result of earlier P300 onset. Regarding the P300 itself, the ERP amplitude interaction of stopping success and tFUS was apparent across all experimental groups (rIFG, S1 and control groups). The P300 amplitude changes were not specific to tFUS changes in performance (i.e. rIFG group), and this ERP peaks after the SSRT. These results point to the timing of the N200/P300 onset latency as a causal and valid marker of inhibitory control.

Is there an interpretation of the P300 amplitude that is bolstered by our results? Notably, our findings contrast with some studies (*Greenhouse and Wessel, 2013*) showing larger P300 amplitudes for successful compared to failed inhibition. In contrast, we found a larger amplitude for unsuccessful trials as also found by others (*Huster et al., 2020*; *Cunillera et al., 2016*). However, it has also been previously that the direction of P300 amplitude differences between successful and failed stopping depends on whether the SSDs are fixed or dynamically staircased (*Waller et al., 2021*). We interpret this confluence of mixed P300 amplitude results as coherent with another interpretation. The P300 amplitude indexes the retrospective evaluation of an outcome expectancy (*Hajcak et al., 2005*), for example How surprising was it that I made a stopping error? This interpretation contrasts with a simple indexing of inhibitory performance, for example Was the stopping outcome successful or did it fail? Our results and others favor the former interpretation. This conclusion is further supported by manipulations of stop signal probability (*Ramautar et al., 2004*) that, in a study like ours, used a fixed set of SSDs. They found that P300 amplitudes were larger when stop signal trials were less likely, and therefore more surprising. Additionally, *Ramautar et al., 2004* found P300 amplitudes were larger in unsuccessful than successful stop trials. Our finding of a tFUS by performance interaction supports this interpretation as well. Speculatively, the P300 amplitude change may reflect the detection of tFUS auditory signals across all groups. If so, the detection of the auditory signal may have increased the subject's retrospective surprise of making a stopping error.

An important consideration is the timing of tFUS effects on stopping in the rIFG group. Effective changes were limited to the longest two SSDs, which we focused our analysis on. Possible explanations for not observing significant tFUS effects for the shorter SSDs are that, even before the Go process has been initiated, the stop/inhibitory process gets initiated, or they get simultaneously initiated but the inhibitory process is automatically prioritized based on recognition of the Stop signal. This would provide less opportunity for tFUS to impact response inhibition, as short SSDs may simply involve 'not going' versus 'inhibiting a Go', which may proceed with a different chronology. This interpretation aligns with the idea that short SSDs violate the independent race model and may invoke different processes (*Bissett et al., 2021*). A further possibility is that tFUS may have a rather delayed (>100ms),

rather than instantaneous impact. For example, tFUS applied directly to cultured cells measured with calcium imaging or GCaMP6f revealed a response delay of approximately 200ms for a 6 w/cm² tFUS intensity (*Yoo et al., 2022*). With increasing intensity, the delay decreased in measurable peak neural activity. This finding, though noting a delay in neural response to tFUS, requires caution for interpretation here for several reasons. First, it reveals neural peak response was intensity dependent, which does not generalize to transcranial application and implies a non-fixed delay. Second, the measure in that study was with calcium measures rather than electrophysiological directly. Calcium imaging has a notable delay, and lower sampling rate than EEG (for example), and could filter the neural response differently. Therefore, while there must be some delay between tFUS and neural response, several more studies in human and animal models across measurement modalities (single units versus EEG) are required to dissociate these issues.

Notably, other studies have attempted to address the gap of a causal connection between rIFG and response inhibition using offline TMS (*Chambers et al., 2006*; *Chambers et al., 2007*; *Sundby et al., 2021*). All studies reported disrupting subjects' ability to inhibit their responses, paired with significant lengthening of SSRT (*Chambers et al., 2006*; *Chambers et al., 2007*; *Sundby et al., 2021*). However, offline TMS-induced behavioral effects are problematic for causal interpretations because TMS (1) offline TMS stimulation cannot deal with temporal confounds of ERPs, and (2) cannot control for carry-over effects between experiment blocks. Using tFUS in our study addresses many of these limitations. Second, we delivered tFUS simultaneously with the Go and the Stop cue and used convolutional GLM modeling to remove overlapping neural effects between Stop and Go ERPs. By doing so, we found response inhibition effects only when tFUS was delivered simultaneously with the Stop cue. Although there is a distinct benefit to tFUS allowing simultaneous stimulation and measurement, care must be taken to establish temporal controls for overlap in macroscale neural measurements, for example EEG. Third, using a spatial control outside of the inhibitory network (primary somatosensory cortex) allowed us to validate the specificity of tFUS behavioral effects to rIFG. More broadly, our study strengthens the evidence for tFUS as a promising technique for neuromodulation to test temporally and spatially precise hypotheses about brain function (*Fini and Tyler, 2017*; *Folloni et al., 2019*; *Reznik et al., 2020*; *Sanguinetti et al., 2020*; *Tufail et al., 2010*; *Tyler et al., 2018*; *Verhagen et al., 2019*; *Yaakub et al., 2023*).

## Materials and methods

### Participants

Healthy adult human volunteers (n=63) were randomly assigned to one of three experimental groups. The main experimental group received transcranial focused ultrasound (tFUS) stimulation to the right inferior frontal gyrus (rIFG) (n=25; 19 males, mean age = 24.1 years, SD = 3.2 years). A second group received stimulation to the ipsilateral somatosensory cortex (n=23; 15 males, mean age = 22.4 years, SD = 3.3 years) and was used as the cortical site active control group (S1). A third group received sham stimulation near the right temple (n=15; 8 males, mean age = 24.2 years, SD = 2.8 years) and was used as a control for possible auditory effects of tFUS modulation (sham rIFG). All individuals were right-handed (self-reported) and received financial compensation for participation in the study. Before enrollment, each subject was screened for neurological disorders and a history of epilepsy, stroke, or brain injury. A neurologist from Barrow Neurological Institute (Phoenix, AZ) screened all subjects' T1 MRIs and cleared them before study participation.

### Behavioral task and transcranial focused ultrasound design

Response inhibition was assessed using the Stop-Signal Task involving both 'Go' and 'Stop' trials (*Figure 1*) programmed in Opensesame (*Mathôt et al., 2012*). Each trial started with a central fixation cross. In every trial, fixations were replaced by a green 'Go' circle (3° x 3° visual angle) after an exponentially distributed time interval (range: 350–650ms; mean: 500ms; standard deviation: 50ms). Subjects were instructed 'to press the 'up' key when detecting the Go circle' (*Figure 1A*). In 'Go' trials (rows 1–2, *Figure 1A*), the circle vanished either after the subject's response or 800ms elapsed. In 'Stop' trials (rows 3–5, *Figure 1A*), the Stop signal was a red square that appeared around the green circle. If the subject successfully inhibited their response with respect to the Stop cue within 800ms, the red square was extinguished, and the trial was considered a successful inhibition. The time

required to inhibit a response following the Stop signal is defined as SSRT (see below). Timing of the Stop cue relative to the Go cue, that is the stop signal delay (SSD), was presented at one of four fixed, but subject-specific SSDs. The SSDs were chosen by having each subject perform a practice block of 50 Go trials to determine their baseline Go reaction time (RT). After this block, the 4 SSD levels were set to 15, 35, 75, and 95% of the mean Go RT. These SSDs were fixed throughout the experimental session and were presented in a random order across Stop trials. All trials were separated by a 2 s inter-trial interval ±300ms random jitter.

We delivered Transcranial Focused Ultrasound (tFUS) simultaneously with the Go signal during Go trials. During stop trials, there were two separate tFUS conditions. In one set, tFUS was delivered simultaneously with the Go signal, and in the other set it was delivered simultaneously with the stop signal (*Figure 1B*); simultaneous here means that tFUS was delivered at time = 0ms – time synched to either the Go signal or the Stop signal. This mixture of Go, Stop and tFUS delivery factors generated 5 trial types (*Figure 1A*). The first two consisted of Go trials with no tFUS or with tFUS time-locked to the Go signal (No-tFUS and Go-tFUS trials, respectively; rows 1–2, *Figure 1A*). The other three trial types consisted of Stop trials with no tFUS, and tFUS time-locked to either the Go or Stop signal (No-tFUS, Go-tFUS, and Stop-tFUS trials, respectively; rows 3–5, *Figure 1A*). tFUS delivery for Stop trials was evenly distributed across the 4 SSD levels. The overall probability of the occurrence of a Stop trial was 44% of all trials. This proportion of trials accommodates the need for a sufficiently large number of Stop trials to examine tFUS effects on Stop trials across all SSD levels while enabling a more frequent occurrence of Go than Stop trials (56%; the percentage of each trial type is shown in *Figure 1A*).

Each experimental session consisted of 1200 trials distributed across 12 blocks of 100 trials each. Blocks were segmented into stimulation/no-stimulation and no-stimulation blocks, the former consisting of trials with and without tFUS, and the latter consisting of trials with no tFUS. Trial types (Go and Stop trials) were randomly distributed throughout the experiment. We chose a blocked design to mitigate possible carry-over effects of tFUS across trials. By using two control groups (S1 and sham rIFG), we could determine the extent to which behavioral and/or neural responses associated with tFUS were specific to the target site (rIFG). The rationale for the two control conditions is provided in detail below.

## EEG and structural imaging acquisition
### EEG recording
EEG was recorded using a 64-channel ActiCap system (BrainVision, Morrisville, NC), with a 10–20 layout. Data was recorded at a sampling rate of 5 kHz, with 0.1 µV resolution and bandpass filter of 0.1–100 Hz. Impedances were kept <5 kΩ. Online recordings utilized a ground at AFz and left mastoid reference. At the beginning of each session, electrode layouts with respect to each individual's head shape were registered using the left and right preauricular, and nasion as fiducial landmarks. This allowed for later co-registration with each individual's T1 structural MRI scan and for source-localized analysis (see below).

### Structural MRI (T1)
For guiding tFUS neuronavigation and co-registering EEG electrode placement for source analysis and modeling, we obtained a structural T1 MRI scan for each participant. T1 volumes were collected using a 3D MPRAGE sequence (TR = 2300ms, TE = 4.5ms, 1x1 x 1.1 mm$^3$ voxels, field of view 240x256 mm$^2$, 180 sagittal slices) in a Philips Ingenia 3T scanner with a 32-channel head coil. Brainsuite was used to process T1s, which included cortical extraction sequence and a surface label-registration procedure with the BCI-DNI atlas. After labeling, we checked the locations and created a mask of either pars opercularis (rIFG group) or the centroid of ipsilateral S1 (S1 group). This volume labeling and mask creation procedure were used for guiding tFUS target identification.

## tFUS targeting, setup and parameters
A BrainSight neuronavigation system (Rogue industries) along with subjects' T1 scans were used to guide the placement of the focused ultrasound transducer beam profile for stimulation. This was done separately with respect to each individual's neuroanatomy and mask created from T1 scans. The first step involved creating a subject-specific mask from cortical atlas registration and projecting it into the Montreal Neurologic Institute (MNI) coordinate system. When planning the tFUS target, we

considered both MNI coordinates and individual anatomy. For example, meta-analysis studies have shown specific activation of the pars opercularis (around x=48, y=16, z=18) for contrasts of successful inhibition versus Go trials and successful versus failed inhibition trials (*Chikazoe et al., 2009*; *Levy and Wagner, 2011*). For the rIFG group, we first identified the pars opercularis MNI coordinates. During target planning, we confirmed the coordinates were inside the anatomical region of pars opercularis. We visually confirmed each subject's pars opercularis tFUS target was rostral to the inferior precentral sulcus and dorsal to the sylvian fissure, and ventral to the inferior frontal sulcus. For the S1 group, tFUS was targeted near MNI coordinates of x=-43, y=-29, z=54 and within the left post-central gyrus.

Before tFUS transducer setup, neuronavigation registered subjects' T1 scans in virtual space, with their head and the ultrasound transducer in real space. Alignment and cortical registration were performed using nasion, tip of the nose, philtrum, and left and right periauricular notch and tragus as fiducial landmarks. A 3D-printed housing held the tFUS transducer, optical trackers, and silicon spacers (ss-6060 Silicon Solutions, Cuyahoga Falls, OH). Acoustic gel was applied to both the transducer and scalp. We recorded stimulation target coordinates after placing the transducer in target alignment. In the sham rIFG group, we delivered sham tFUS (*Legon et al., 2018*). The transducer pointing away and perpendicular to the scalp, placing the transducer where electrode F8 would have been. This is the same electrode that was also not used during the active TFUS rIFG group and was chosen to approximate the area where the transducer was typically placed in the rIFG experimental group. The logical basis of this sham was that we (a) wanted to emulate the sound that is cranially detectable during active TFUS, and (b) did not want substantial ultrasound energy transmitted. The choice of this sham was codified as producing a detectable sound when placed against the skull.

Here, we clarify the logic of our control groups more thoroughly. We used the sham condition to control for auditory effects, as we found that coupling the transducer to the scalp and placing it perpendicular to the scalp would prevent sonication, but still induced an auditory ringing similar to active tFUS. We reasoned that if an auditory effect drove our behavioral or neural results, then we would find similar changes in inhibition performance effects in the Sham, S1 control, and rIFG group. The S1 control group was used as an active sonication and auditory control to discern the specificity of the rIFG tFUS. Put differently, if the S1 and rIFG group exhibited similar behavioral changes, at minimum, this could result from an auditory artifact specific to active tFUS as both groups would experience similar auditory sensations. Thus, while the literature (*Braun et al., 2020*) remains uncertain about how to best define a sham control for auditory effects, we approached the problem by using both active tFUS (S1 group) and sham tFUS to control for auditory effects. To preface our results, we find support for the specificity of our results in that the joint impact of tFUS in behavior and neural data was limited to the rIFG group and not present in the S1 group or control group.

For the rIFG and S1 groups, we measured the accuracy of stimulation target coordinates by tracking the deviation of the tFUS beam profile from the cortical target throughout the experiment. During the experimental session, we sampled the tFUS transducer spatial target deviation during each break. Accuracy was very high, with an average deviation of ±1.5 mm displacement across all subjects and sessions.

Our tFUS setup and parameters were nearly identical to those used by *Legon et al., 2014*. Briefly, we used a single-element tFUS transducer with a center frequency of 0.5 MHz, a focal depth of 30 mm, a lateral spatial resolution of 4.5 mm, and an axial spatial resolution of 18 mm (Blatek). tFUS waveforms were generated using a two-channel, 2 MHz function generator (BK Precision). The system operated by channel 1 produced a pulse repetition frequency (PRF) of 1.0 kHz. Channel 1 also triggered channel 2, which produced short bursts at the 0.5 MHz acoustic frequency. This produced an ultrasound waveform with a carrier frequency of 0.5 MHz, PRF of 1.0 kHz, and a duty cycle of 24%. Each stimulation duration was 0.5 s. Transducer power was driven by output from a 40 W linear RF amplifier (E&I 240 L; Electronics and Innovation).

## Computational simulation and validation of tFUS propagation

We quantified peak pressure amplitude, peak intensity and accuracy of the tFUS beam distribution delivered to rIFG using the pseudospectral simulation method in K-wave (*Treeby and Cox, 2010*). Reference peak pressure planes for the simulations were derived from previous data (*Legon et al., 2014*). Simulation parameters were first validated by simulating the transducer in water to compare the simulation results with those from previous water tank tests (*Legon et al., 2014*). The maximum

pressure plane at the 30 mm focus was used as a source input pressure for the transducer during the simulation. The transducer was modeled to have a 30 mm radius of curvature. For water simulations, we used a homogenous medium of water density (1000 kg/m³) and speed of sound (1482 m/s). We created a computational grid (270x280 x 231) with 1 mm spacing. The points per wavelength were 6, Courant–Friedrichs–Lewy = 0.1, and simulation time was set to 6 pulses (duration = 250 μs) to ensure simulation stability.

For simulating transcranial ultrasound stimulation, we extracted 3-dimensional maps of the skull from a CT (1 mm resolution) and brain from T1 MRI scans (1 mm resolution) from three preoperative patients at Barrow Neurological Institute. The MRI and CT were both co-registered and normalized to the MNI space in SPM12. To mimic our approach of tFUS targeting used in the experiments, we surface-registered the gray matter volume to the BCI-DNI atlas and identified the centroid of pars opercularis. The average stimulation location for these three subjects was x=48, y=18, and z=6. This allowed us to map from world coordinates of the scan to MNI coordinates of the target. *Figure 1— figure supplement 1A* shows T1 and scalp from one subject, together with renderings of the transducer housing, the pars opercularis mask, and the tFUS target applied to all MRIs from the rIFG group. *Figure 1—figure supplement 1B* shows side views of non-normalized T1s, pars opercularis masks, and the tFUS targets (red dots) for four subjects (*Figure 1—figure supplement 1B*). Conversion from Hounsfield units in the CT to sound speed and density was done using the relations described in *Aubry et al., 2003*. All skull materials were set using these parameters, while other tissues were treated as homogeneous with parameters set to that of water. Attenuation was modeled as a power law with a $\beta$=0.5 while absorption was modeled with $b$=1.08 (*Treeby and Cox, 2010*).

To assess transcranial stimulation accuracy, the simulated transcranial transmission was compared against simulations of tFUS transmission through water. Differences between these simulations show the estimated effect of any power absorption and change in acoustic profile after skull transmission. Numerical simulation parameters (see above) were derived to ensure the water simulation here matched the water tank results from a previous study using the same transducer and tFUS experimental parameters (*Legon et al., 2014*). Simulation of ultrasound through water predicted a maximum pressure ($P_{max}$) of 0.82 MPa, spatial peak pulse average intensity ($I_{SPPA}$) of 22.43 W/cm² at the focus, spatial peak temporal average intensity ($I_{SPTA}$) of 5.38 W/cm² at the focus, lateral full-width at half maximum of the maximum pressure (FWHM) of 3.56 mm and mechanical index (MI) of 1.15. (*Figure 1—figure supplement 1C*).

Comparison of simulations and previous water tank data (*Legon et al., 2014*) using the same transducer and experimental tFUS parameters indicated a 97% match of pressure/intensity at the focus taken over a 5 mm³ voxel section in all three planes at the focus. Next, modeling of transcranial transmission predicted $P_{max}$ 0.54, $I_{SPPA}$ 10.01 W/cm² and $I_{SPTA}$ 2.40 W/cm², which is the intensity range of non-thermal neuromodulation and that exhibits nearly instantaneous measurable effects on EEG (*Legon et al., 2014*). Comparing the water and transcranial simulation, accuracy was assessed by comparing shifts in peak pressure. Skull transmission compared to water was shifted 1.25 mm laterally and had a lateral beam profile full-width half maximum of 6.13 mm (*Figure 1—figure supplement 1C*). These transcranial simulations indicate high spatial precision, with >95% of pressure or energy (kPa) being constrained to pars opercularis in the rIFG group (*Figure 1—figure supplement 1D*). We note that *Legon et al., 2014* used acoustic gel only, whereas we used a silicone puck and acoustic gel. Therefore, to accommodate for tFUS attenuation caused by the silicone puck, we also simulated the ultrasound environment by simulating the silicone puck interposed between the transducer and the skull in a water environment. As the acoustic properties of our silicone puck were unavailable, we simulated two densities corresponding to the minimum and maximum possible values of undoped silicone (1200 and 1550 kg/m³). All the simulation results are presented in *Table 2*.

## Statistical analysis

### Behavioral variables

Our behavioral analyses focused on the following variables: Go trial reaction time (Go RT), percentage of failed inhibition responses given a stop signal was present (P(respond|signal)), signal response reaction time (failed inhibitions; SRRT), and SSRT. We next describe our steps for estimation of each of the relevant variables.

**Table 2.** tFUS simulation results showing intensity measures ($P_{max}$, $I_{SPPA}$, and $I_{SPTA}$), lateral target focality (FWHM) and mechanical index (MI) measures.

Results are displayed for four different simulations, including only water, transcranial without silicone puck, and 2 simulations with silicone pucks of different densities.

| Simulation | $P_{max}$ (in MPa) | $I_{SPPA}$ (W/cm²) | $I_{SPTA}$ (W/cm²) | FWHM (mm) | MI |
|---|---|---|---|---|---|
| Only water | 0.82 | 22.43 | 5.38 | 3.56 | 1.15 |
| Transcranial without silicone puck | 0.54 | 10.01 | 2.40 | 6.13 | 0.77 |
| With silicone puck (density estimated at 1200 kg/m³) | 0.53 | 9.82 | 2.35 | 5.25 | 0.76 |
| With silicone puck (density estimated at 1550 kg/m³) | 0.52 | 9.37 | 2.25 | 5.27 | 0.74 |

The SSRT was estimated using a hierarchical Bayesian parametric approach (*Matzke et al., 2013*) that estimates the distribution of SSRTs while assuming an ex-gaussian parametric form. We chose this approach as *Matzke et al., 2013* showed that it performs well even when there are only a few trials available per SSD level. This SSRT estimation procedure was run separately per subject and group (rIFG, S1, and Sham rIFG). Within each subject's fit, trial types (No-tFUS Stop trials, Go-tFUS Stop trials, and Stop-tFUS Stop trials; *Figure 1*) were combined to create a hierarchical, within-subject model estimation of SSRT. For these analyses, we used Go RTs combined from Go trials with and without tFUS because stimulation did not alter the Go RT (shown in Results).

In our SSRT estimation and the rest of our analyses, we only included the two longest SSDs (75% and 95%). This choice is based on conforming to the assumptions of the underlying estimation methods, the independent race model of inhibition, and to avoid bias in the estimation of key parameters (i.e. SSRT) induced by violating model assumptions. Specifically, the above method of SSRT estimation and nearly all others (e.g. *Verbruggen et al., 2019*) relies on independence assumptions put forth by the independent race model of inhibition (*Bissett et al., 2021*; *Logan and Cowan, 1984*; *Matzke et al., 2013*; *Verbruggen et al., 2019*). An important assumption for the validity of the method is context independence (*Logan and Cowan, 1984*). Context independence assumes the finishing times of the Go RT distribution are unaltered by the presence of the stop signal. The context-independence assumption predicts that SRRT (across SSD levels) should be shorter than the Go RTs that occur without a stop signal. This is because the fastest Go responses during a stop trial are the ones that escape inhibition (*Logan and Cowan, 1984*). Notably, several studies have shown that violations measured by SRRT being longer than Go RT tend to occur at short SSDs (<200ms; *Colonius et al., 2001*; *Logan and Cowan, 1984*). *Bissett et al., 2021* noted these studies were severely underpowered. Important for the present study is the work by *Bissett et al., 2021* who showed that meaningful violations were limited to SSDs less than 100–150ms. In our study, the 75% and 95% SSDs were generally above this range where violations are presumed to occur, leading us to focus our SSRT estimation and analyses on these 2 levels. Across all groups and subjects, the mean 75% and 95% SSD were 205ms (SD:±31ms) and 260ms (±38ms).

To analyze the probability of failing to inhibit responses, that is P(respond|signal), across the factors of SSD, tFUS and Group, we used a mixed-design ANOVA. In the model, SSD had 2-factor levels (75% and 95%), tFUS had three levels (None, Stop, Go), and between-subjects Group factor had three levels (rIFG, S1, Control). The tFUS level of None means no tFUS was applied, the Stop level refers to tFUS applied simultaneously with the Stop signal, and Go refers to tFUS applied simultaneously with the Go signal (during Stop trials).

To analyze Go RT and SRRT, we first extracted their mean and standard deviations by fitting both RT types with an ex-gaussian distribution using maximum likelihood (*Lacouture and Cousineau, 2008*). This choice was made to ensure alignment between the SSRT estimation procedure and the known ex-Gaussian shape of RT distributions (*Lacouture and Cousineau, 2008*). With the SRRT, we estimated the distributional moments using the combined SRRTs from the 75% and 95% SSDs.

Both the Go RT and SRRT were also analyzed with mixed-design ANOVAs, albeit with different designs. ANOVAs for Go RT included the factors of Group (3 levels) and tFUS (None and GO). Note that the Go tFUS here means tFUS applied simultaneously with the Go signal in Go-only trials where

there was no stop signal. This analysis was performed to assess whether tFUS to a specific brain area (rIFG or S1) or auditory effects alone could alter Go responses. The mixed-design ANOVA for SRRT was used to assess whether Go processes that escaped inhibition were altered by tFUS or other confounds, for example auditory stimuli. If these effects were prevalent, even if brain area specific, they would suggest that tFUS impacted the Go and not the Stop process during Stop trials. This ANOVA had factors of tFUS (None, Stop, Go) and Group. For both Go RT and SRRT, we used identical ANOVAs to separately analyze their means and variability (standard deviation).

## EEG pre-processing

Continuous EEG data were first down-sampled to 250 Hz, then high-pass filtered (0.1 Hz) to remove baseline drift. All channels with visually identifiable noise artifacts for more than 25% of the recording length were then removed. In addition, for each of the active stimulation groups (rIFG and S1), the cortical sites of rIFG and S1 were close to the F8 and CP4 electrodes. Therefore, these electrodes could not be used for EEG recording in their groups. Continuous EEG segments were then visually identified and removed where channels exhibited stereotypical muscle and movement activity artifacts (removed portions <8% of recording lengths across participants). Out of all participants, five subjects were excluded (2 from the S1 group and 3 from the rIFG group) from analyses due to EEG recording issues and noise (impedance >25 kΩ across channels). To remove other artifactual data segments, we applied artifact subspace reconstruction (ASR) to remove data identified as artifacts. This function was implemented using the 'clean_rawdata()' function in EEGlab and was used to identify noisy segments as >16 standard deviations above the mean. This threshold was chosen based on previous method sensitivity studies (*Chang et al., 2018*). Briefly, ASR effectively uses a sliding window form of principal components analysis to identify large outlier signals and find non-stationary outliers. By removing these non-stationary outliers, ASR has been shown to improve the later decomposition of EEG using independent components analysis (ICA) in terms of the quality of dipolar maps returned by ICA (*Delorme et al., 2012*). After removing the segments identified by ASR, all previously removed electrodes were interpolated and the data were common average referenced to ensure a zero-sum voltage across channels.

Independent components analysis (ICA) was then used to obtain canonical activity maps across electrodes for further analysis, a standard approach in inhibitory control electrophysiology (e.g. *Huster et al., 2020*; *Wessel and Aron, 2015*; *Wessel et al., 2019*). Each subject's data were processed with the Adaptive Mixture of ICA (AMICA; *Palmer et al., 2008*) approach using one mixture model. AMICA was chosen because it has been shown to better separate data into independent components with dipolar spatial maps (*Delorme and Makeig, 2004*).

Because our hypotheses are a priori aimed at determining the link in the N200/P300 ERP to inhibition and tFUS, we next selected a singular ICA component, for each subject, that had the desired ERPs (N200/P300) putatively linked to inhibitory control (*Huster et al., 2020*; *Wessel et al., 2013*). A plethora of studies using this approach have shown that ICs with a medial-frontal topography (centered around electrodes Cz and Fcz) exhibit the N200/P300 and are modulated by successful stopping during inhibitory control tasks (*Huster et al., 2020*; *Wagner et al., 2018*; *Wessel and Aron, 2015*).

To obtain desired ICs, we used a similar approach to previous work analyzing ICA components for electrophysiological markers during inhibition (*Wagner et al., 2018*; *Wessel and Aron, 2015*). Singular equivalent current dipoles were fit for each IC scalp map using DipFit 2.2 in the EEGLAB toolbox. Any IC with a dipole residual variance of less than 15% for predicting the scalp map was removed as these are typically considered to be signals of non-brain activity origin. The remaining ICs across subjects were then clustered using a K-medoids algorithm. We chose a K-medoids approach for clustering as its similarity metric is more robust to noisy outliers, as compared to K-means approaches that rely on centroids (*Velmurugan and Santhanam, 2010*). To avoid double dipping in our analysis of ERPs, we only used the features of 3D (X, Y, Z) dipole position and IC weight spatial maps. We optimized the selection of the number of clusters using the adjusted rand index (ARI), which measures the stability of the clustering results over multiple runs of the K-medoids algorithm. ARI estimates how similar each cluster is across the runs (from 0 to 1 being identical clusters). We found a maximum ARI of 0.82 for 11 clusters. To select a cluster of ICs for analyzing the N200/P300, we used a previously-employed criterion (*Wessel, 2016*) of requiring it to have: (1) a fronto-radial spatial map topography and

(2) a maximal IC weight at 1 of the Fcz, Fz, Cz, Fc1, Fc2 electrodes. For each cluster, some subjects had multiple ICs. In that case, we retained only the subject IC that maximally correlated with the cluster spatial template and had a dipole with a smaller distance from the IC spatial map centroid position from the cluster averaged dipole position.

## EEG statistics

### Deconvolution GLM for regression ERPs (rERPs) and separating Stop from Go response ERPs

For the ERP analysis using the retained subject medial frontal IC components (see above), we first performed a general linear model (GLM) with deconvolution at the individual subject level using the 'unfold' toolbox in Matlab (*Ehinger and Dimigen, 2019*). We used a deconvolution approach to separate temporally adjacent ERP activity from the Go response that overlaps with the Stop signal. This approach creates regression-based ERPs (rERPs). This is helpful because the different SSD levels render different amounts of the Go process that have evolved since the onset of the Go signal. Put differently, the Go-related activity will have evolved for a longer time as a function of increasing SSD. Therefore, the extent and amplitude of overlaps between Go and Stop ERPs will differ according to SSD. By modeling the Go process ERP in a GLM and distinguishing it from the Stop ERP activity, we are able to capture a less confounded Stop ERP by minimizing the differences in Stop ERP activity that would be related to differences in Go RT caused by different SSDs. In addition, the deconvolution approach models the overlap directly rather than making distributional assumptions about the latent Go and Stop reaction times (*Mattia et al., 2012*). For a thorough description of how a deconvolution GLM works, the reader is referred to the 'unfold' toolbox (*Ehinger and Dimigen, 2019*) or other sources for general explanations of first-level electrophysiological analysis using GLMs (*Litvak et al., 2013*).

To obtain each subject's rERP, we modeled them (GLM) using two events, the Go stimulus event and the Stop signal event. To obtain a full design set of Stop signal ERPs, we modeled the Stop ERP by crossing all factors of successful and unsuccessful stopping (SS minus US), SSD, and tFUS event-aligned conditions (None, Stop, and Go). The Go-signal ERP is considered an overlapping confound here, and only its intercept was modeled (y~1). Intercept-only modeling allows for removing overlapping additive effects of the Go with the Stop activity (*Ehinger and Dimigen, 2019*). The stop signal rERPs had a full model in Wilkinson Notation as follows:

$$\text{rERP(stopsignal)} = y \sim 1 + (\text{SSD})^*(\text{SS} - \text{US})^*(\text{tFUS}).$$

All events and predictors were fit using a cubic-spline basis set (40 splines). The design matrix of spline bases was time expanded in a window of –500 ms to 1000 ms around each event of interest. The model was fitted using the 'maximum-likelihood' function in Matlab. Before fitting each model, continuous artifactual segments were rejected using the events found from the ASR cleaning method (see EEG pre-processing).

ERPs for each of the Stop signal conditions were then composed from the rERPs by using a contrast code. At this level of ERP composition, we only considered the two highest SSDs to follow the same logic as in the behavioral analysis, that is to minimize bias from SSDs (<100–150ms) that would, otherwise, not conform to the independence assumptions of the race model (see above for rationale). In addition, we only consider the rERPs related to the successful and unsuccessful stopping during the No- and Stop-tFUS conditions. We make this choice because Go-related activity – and therefore Go-tFUS activity – is effectively deconvolved out, and to foreshadow the behavior, we found no impact of Go-tFUS on behavior.

## ERP statistics

Our predictions about the relevant medial frontal ERP (N200/P300) components and their timing (P300 onset) were established based on a long history of previous studies. Therefore, we use a circumscribed approach of analyzing the mean amplitude around the peaks of the ERP components using pre-defined time windows. In all these analyses, we used the ERP activity defined from the first-level GLM analysis of each IC. First, we defined time windows around canonical components that appear in inhibition tasks, including a P100 (25–150ms window), N200 (165–255ms window), and P300 (280–450ms window). For each subject and each condition-specific rERP for successful and

unsuccessful stopping, and tFUS (No- or Stop-), we estimated the time-index of either the maximum (P100 and P300) or minimum (N200) amplitude within their respective windows. Next, we computed the weighted amplitude mean by convolving a Gaussian window of size ±30ms (7 points) centered at the peak. We used mean amplitude as its linearity provides comparability across conditions, as well as being less sensitive to noise, trial numbers, and timing variability than the peak alone (see *Luck and Gaspelin, 2017* for a thorough explanation).

To quantify these ERP means, we used a mixed-design ANOVA with a two-level factor describing inhibition success (successful stopping: SS, or unsuccessful stopping: US), a two-level tFUS factor (No- or Stop-), and a three-level between-group factor (rIFG, S1, Control). As multiple ERPs were tested, we used a false-discovery rate (FDR) correction of 0.05 within each main effect or interaction. For example, we computed the FDR-corrected p-values across the effect for SS and US conditions.

## Deriving P300 onset latency and regression with change in SSRT

Recent work has indicated that frontocentral (ERP) P300 onset latency is related to the speed of successful inhibition (SSRT) across subjects (*Wessel and Aron, 2015*). Therefore, we regressed the between-subject changes in SSRT as a function of the P300 latency change between No-tFUS and Stop-tFUS or Go-tFUS in successful stop trials. To achieve this, we computed the shift in P300 onset crossings between the No-tFUS and other tFUS conditions. We first found the zero-crossing around the N200 (using the same time windows for ERP amplitude analysis) for each subject's mean fronto-central ERP. We then computed the dynamic time warping distance (DTW) between the No-tFUS and either Stop- or Go-tFUS ERP waveform in a time-window of ±50ms around the zero-crossing found in the No-tFUS waveform; we opted to use DTW to compute P300 onset latency differences because previous work has shown its superiority to other latency computation methods (*Zoumpoulaki et al., 2015*). Therefore, the DTW provides a temporal distance metric for P300 onset latency differences between tFUS conditions and No-tFUS ERPs.

Using the P300 onset differences, we performed a difference-in-differences analysis (*Cunningham, 2021*). Specifically, we regressed the onset differences by individual subject differences in SSRT between conditions. For example, the difference in latency between No-tFUS and Stop-tFUS P300 was regressed against the difference in SSRT between the No-tFUS and Stop-tFUS conditions (and the same for Go-tFUS compared to No-tFUS). To ascertain measures of the probability of the regression slope being different from zero, we used a Bayesian regression with a Hamiltonian Monte-Carlo (MCMC) sampler in Matlab. This approach has the added benefit of being shown to outperform standard bivariate correlations in small sample sizes as exists in the group sizes here (n<25 per group; *Fosdick and Raftery, 2012*). Doing so allowed us to ascertain the posterior probability using the 95% posterior credible intervals of the regression slope and their overlap with 0. All prior means on the intercept, slope and log variance term were set to 0, assuming no a priori relationship between the difference measures. For the sampler, we used a burn-in of 1000 samples, 4 chains, and 10,000 samples in total, with a chain step size of 10.

## Code availability

Behavioral and ERP Analysis scripts for reproducing figures are available at: https://github.com/Just-FineNeuro/tFUS_Inhibition_Elife (copy archived at *Fine, 2024*).

---

## Additional information

### Funding

No external funding was received for this work

### Author contributions

Justin M Fine, Conceptualization, Data curation, Formal analysis, Supervision, Validation, Visualization, Methodology, Writing – original draft, Writing – review and editing; Archana S Mysore, Data curation, Writing – review and editing; Maria E Fini, Conceptualization, Data curation, Methodology, Writing – original draft; William J Tyler, Conceptualization, Resources, Supervision, Writing – review and editing;

Marco Santello, Conceptualization, Resources, Supervision, Funding acquisition, Writing – original draft, Writing – review and editing

**Author ORCIDs**
Justin M Fine ⬥ https://orcid.org/0000-0003-2378-6854

**Ethics**
Informed consent, safety checklists, and right to publish data were obtained from each participant in the study. All study procedures were approved and performed in accordance with the institutional review board at Arizona State University (STUDY00006050).

**Decision letter and Author response**
Decision letter https://doi.org/10.7554/eLife.86190.sa1
Author response https://doi.org/10.7554/eLife.86190.sa2

## Additional files

**Supplementary files**
• MDAR checklist

**Data availability**
The human behavioral and EEG ERP datasets for reproducing figures are publicly available. The datasets are publicly available on Dryad: https://doi.org/10.5061/dryad.sj3tx968j.

The following dataset was generated:

| Author(s) | Year | Dataset title | Dataset URL | Database and Identifier |
| --- | --- | --- | --- | --- |
| Fine JM, Fini M, Mysore AS, Tyler WJ, Santello M | 2023 | Transcranial focused ultrasound to rIFG improves response inhibition through modulation of the P300 onset latency | https://doi.org/10.5061/dryad.sj3tx968j | Dryad Digital Repository, 10.5061/dryad.sj3tx968j |

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
