## [Editor Report]

This study presents a valuable finding on the causal contribution of the inferior frontal gyrus (IFG) in behavioral control. State-of-the-art transcranial ultrasonic stimulation in combination with EEG is used to stimulate the IFG and find changes in speed and accuracy in a stop-signal task. This convincing work will be of interest to a wide range of basic neuroscientists.

---

## [Decision Letter]

**Decision letter after peer review:**

Thank you for submitting your article "Transcranial focused ultrasound to rIFG improves response inhibition through modulation of the P300 onset latency" for consideration by *eLife*. Your article has been reviewed by 2 peer reviewers, and the evaluation has been overseen by a Reviewing Editor and Michael Frank as the Senior Editor. The following individual involved in the review of your submission has agreed to reveal their identity: Lennart Verhagen (Reviewer #2).

Essential revisions:

1. The use of silicone (without doping) is not recommended as a coupling medium. While its sound velocity and attenuation are appropriate, the density of undoped silicone is significantly different from human tissue Z 0.97 vs. 1.55. As such, the reported pressure/intensity in the head is unlikely to be accurate and significantly lower than reported. I would suggest you empirically test this medium in a tank or include it in your models.

2. Are the W/cm^ values reported Isppa or Ispta?

3. There are multiple missing references. Matzke, Wagner, Lacoutoure … this is sloppy and seriously detracts from the manuscript.

4. The EEG pre-processing is insufficient. There is mention of interpolation several times but no details. Data is said to be re-referenced 3 separate times … is this correct? If so, why?

5. Citations are needed for many of the methods for example the use of artifact subspace reconstruction, k-medoids, dynamic time warping, etc. You may know who and why these methods are used but many others will not. There is a line in the methods that this is a "standard approach in inhibitory control electrophysiology" – cite something.

There is mention of the approach of Mattia. What is it?

6. The data in figure 3 does not appear to be the data of the main logistic regression analysis but is the crux of the paper. The ANOVA before the figure refers to Figure 3 but so also does the ANOVA after Figure 3. Which is it?

7. The data in Figure 3A are from the ANOVA from the logistic regression? I don't think so as these are probabilities and not log odds but are referred to as such. Please confirm. Also, there is another ANOVA done separately for the different SSDs? It is also not clear what data, and how many trials went into the analysis (especially the regression). There is mention of estimating data points. Why was this done? What happened to the actual data?

8. A table outlining the data; means etc. plus the results of all tests would be very helpful in sorting through all the data.

9. Finally, the interpretation of the ERP data in Figure 4 looks to be all descriptive. What tests were done to determine latency or peak differences? How were these quantified if at all? Be sure to cite for the use of permutation statistics and were these done only in sensor space but not temporally across the waveforms? How was latency/onset determined and statistically tested?

10. I think adding the direct comparison between rIFG-tFUS and S1-tFUS and between rIFG-tFUS and sham-tFUS to figures 3 and 4 would strengthen the study.

11. It would be wonderful to learn more about the sham condition. From the methods I couldn't work out how this was performed, how the transducer was placed, whether stimulation was applied or only a sound, or how exactly this condition controlled for possible auditory effects.

12. A figure with the exact trial timing would be very helpful to understand the timing of the go cue, SSD, the exact timing of the tFUS relative to the go and/or stop signal, the temporal overlap between tFUS and SSRT and ERP, etc.

13. Was the tFUS at stop delivered 100 ms before the stop signal (as in Legon 2014, t=-100ms), or at the time of the stop signal (t=0ms)?

14. Did the stop signal (red square) appear at random times (as suggested in the caption of figure 1), or at one of four discrete SSDs (as suggested by the methods section)?

15. The description of the tFUS methods would benefit from 1) a clearer specification of the intensity together with the other stimulation parameters, and 2) an estimation of indices relevant for safety, such as the MI and estimated thermal rise in the skull and brain or TIC.

16. You report that tFUS improved stopping behaviour. This was only the case for the SSD at 95% of baseline RT, right? It seems that with SSD at 65% of baseRT the performance was impaired (Figure 3A). Although in the text on page 21, the opposing effects at 65% and 95% are together described as '[…] inhibitory performance appear greater at longer SSDs'.

17. On page 17, I didn't quite understand how the ERPs corresponded to the 85% and 105% of mean go RT, while the SSDs corresponded to the 65% and 95% of the baseline go RT. Is this because there was a difference between baseline go RT and task go RT?

18. The EEG analyses might benefit from quantified statistical analyses. Cluster-based permutation tests are described in the methods, and three contrasts are mentioned, but it seems none are reported in the Results section.

19. In general, the methods section, especially the description of the statistical inference, could benefit from more clarity and more rigorous and consistent descriptions of conditions and parameters.

20. I would recommend using the same y-axis range across the panels of figure 4B. This will allow a direct comparison of the ERPs between no-tFUS and stop-tFUS conditions.

21. The discussion might benefit from considering the tFUS timing in relation to the SSRT. How much tFUS was delivered before the SSRT was reached? Especially considering that the effect was only present at SSD = 95%. How does this relate to the known delay between tFUS onset and modulation of spiking activity (e.g. 50-100ms delay)?

22. The discussion mentions that TMS has a resolution in the order of a few millimeters and TUS of about 1-2mm. This seems similar, and both are somewhat optimistic. The lateral FWHM of the TUS beam was already 5mm, and the axial/longitudinal FWHM is probably in the order of centimeters.

23. Do you have any ideas on the neurophysiological mechanism through which tFUS led to improved stopping behaviour (or impaired at 65% SSD)?

24. Some conclusions from the discussion do not seem to be strongly supported by evidence. For example, I did not fully understand how you conclude that rIFG tFUS is not involved in P300 amplitude modulation. Also, the conclusion on N200 amplitude relation to response inhibition outcome seems to be based on a visual comparison. Did I miss something? And the same for the conclusion that N100 was modulated in the no-tFUS, but not in other tFUS conditions? Was this also a visual comparison?

*Reviewer #1 (Recommendations for the authors):*

I found this manuscript a real challenge to read and interpret. The background and premise were solid and the use of online tFUS to modulate the rIFG to see what this does to stop-signal behavior and ERPs was solid. Once I hit the methods/results however I got lost. Below are a few issues that I feel need to be addressed before publication.

Methods

The use of silicone (without doping) is not recommended as a coupling medium. While its sound velocity and attenuation are appropriate, the density of undoped silicone is significantly different from human tissue Z 0.97 vs. 1.55. As such, the reported pressure/intensity in the head is unlikely to be accurate and significantly lower than reported. I would suggest you empirically test this medium in a tank or include it in your models.

Are the W/cm^ values reported Isppa or Ispta?

There are multiple missing references. Matzke, Wagner, Lacoutoure … this is sloppy and seriously detracts from the manuscript.

The EEG pre-processing is insufficient. There is mention of interpolation several times but no details. Data is said to be re-referenced 3 separate times … is this correct? If so, why? Citations are needed for many of the methods for example the use of artifact subspace reconstruction, k-medoids, dynamic time warping, etc. You may know who and why these methods are used but many others will not. There is a line in the methods that this is a "standard approach in inhibitory control electrophysiology" – cite something.

There is mention of the approach of Mattia. What is it?

– The data in figure 3 does not appear to be the data of the main logistic regression analysis but is the crux of the paper. The ANOVA before the figure refers to Figure 3 but so also does the ANOVA after Figure 3. Which is it?

– The data in Figure 3A is from the ANOVA from the logistic regression. I don't think so as these are probabilities and not log odds but are referred to as such. Please confirm. Also, there is another ANOVA done separately for the different SSDs? It is also not clear what data, and how many trials went into the analysis (especially the regression). There is mention of estimating data points. Why was this done? What happened to the actual data?

– A table outlining the data; means etc. plus the results of all tests would be very helpful in sorting through all the data.

– Finally, the interpretation of the ERP data in Figure 4 looks to be all descriptive. What tests were done to determine latency or peak differences? How were these quantified if at all? Be sure to cite for the use of permutation statistics and were these done only in sensor space but not temporally across the waveforms? How was latency/onset determined and statistically tested?

*Reviewer #2 (Recommendations for the authors):*

Fab work. I have been hoping to see this published for a while.

I think adding the direct comparison between rIFG-tFUS and S1-tFUS and between rIFG-tFUS and sham-tFUS to figures 3 and 4 would strengthen the study.

It would be wonderful to learn more about the sham condition. From the methods I couldn't work out how this was performed, how the transducer was placed, whether stimulation was applied or only a sound, or how exactly this condition controlled for possible auditory effects.

A figure with the exact trial timing would be very helpful to understand the timing of the go cue, SSD, exact timing of the tFUS relative to the go and/or stop signal, temporal overlap between tFUS and SSRT and ERP, etc.

Was the tFUS at stop delivered 100 ms before the stop signal (as in Legon 2014, t=-100ms), or at the time of the stop signal (t=0ms)?

Did the stop signal (red square) appear at random times (as suggested in the caption of figure 1), or at one of four discrete SSDs (as suggested by the methods section)?

The description of the tFUS methods would benefit from 1) a clearer specification of the intensity together with the other stimulation parameters, and 2) an estimation of indices relevant for safety, such as the MI and estimated thermal rise in the skull and brain or TIC.

You report that tFUS improved stopping behaviour. This was only the case for the SSD at 95% of baseline RT, right? It seems that with SSD at 65% of baseRT the performance was impaired (Figure 3A). Although in the text on page 21, the opposing effects at 65% and 95% are together described as '[…] inhibitory performance appear greater at longer SSDs'.

On page 17, I didn't quite understand how the ERPs corresponded to the 85% and 105% of mean go RT, while the SSDs corresponded to the 65% and 95% of the baseline go RT. Is this because there was a difference between baseline go RT and task go RT?

The EEG analyses might benefit from quantified statistical analyses. Cluster-based permutation tests are described in the methods, and three contrasts are mentioned, but it seems none are reported in the Results section.

In general, the methods section, especially the description of the statistical inference, could benefit from more clarity and more rigorous and consistent descriptions of conditions and parameters.

I would recommend using the same y-axis range across the panels of figure 4B. This will allow a direct comparison of the ERPs between no-tFUS and stop-tFUS conditions.

The discussion might benefit from considering the tFUS timing in relation to the SSRT. How much tFUS was delivered before the SSRT was reached? Especially considering that the effect was only present at SSD = 95%. How does this relate to the known delay between tFUS onset and modulation of spiking activity (e.g. 50-100ms delay)?

The discussion mentions that TMS has a resolution in the order of a few millimeters and TUS about 1-2mm. This seems similar, and both are somewhat optimistic. The lateral FWHM of the TUS beam was already 5mm, and the axial/longitudinal FWHM is probably in the order of centimeters.

Do you have any ideas on the neurophysiological mechanism through which tFUS led to improved stopping behaviour (or impaired at 65% SSD)?

Some conclusions from the discussion do not seem to be strongly supported by evidence. For example, I did not fully understand how you conclude that rIFG tFUS is not involved in P300 amplitude modulation. Also, the conclusion on N200 amplitude relation to response inhibition outcome seems to be based on a visual comparison. Did I miss something? And the same for the conclusion that N100 was modulated in the no-tFUS, but not in other tFUS conditions? Was this also a visual comparison?

---

## [Author Response]

Essential revisions:Reviewer #1 (Recommendations for the authors):

We are pleased that this reviewer was overall positive about our work, stating that “The background and premise seem solid” and that “the experimental design looks appropriate with good controls”. However, the reviewer raised concerns about the interpretation of our results as well as lack of methodological details and citations. The reviewer also stated: “Despite the fact that there are many statistical tests in the results, there are none for their main conclusions that the P300 latency indexes stop-signal inhibition – this is only descriptive.” We appreciate the constructive criticism and have addressed all of the concerns raised by the reviewer. Below we provide a point-by-point reply to each critique.

MethodsThe use of silicone (without doping) is not recommended as a coupling medium. While its sound velocity and attenuation are appropriate, the density of undoped silicone is significantly different from human tissue Z 0.97 vs. 1.55. As such, the reported pressure/intensity in the head is unlikely to be accurate and significantly lower than reported. I would suggest you empirically test this medium in a tank or include it in your models.

We appreciate the comment. We did not dope the silicone coupling medium for our experiments. Nevertheless, we have addressed the reviewer’s suggestion to include the silicone as a coupling medium in our simulations. We now report our procedures as follows: We note that Legon et al. (2014) used acoustic gel, whereas we used a silicone puck. Therefore, to accommodate for tFUS attenuation caused by the silicone puck, we also simulated the ultrasound environment by simulating the silicone puck interposed between the transducer and the skull in a water environment. As the acoustic properties of our silicone puck were unavailable, we simulated two densities corresponding to the minimum and maximum possible values of undoped silicone (1200 and 1550 kg/m^3^). We report all of the updated intensity values now in Table 1 and Figure 2C.

Are the W/cm^ values reported Isppa or Ispta?

All the simulation values (new and old) of W/cm^2^ are reported in Isppa and Ispta are now reported in Table 1 in the section “Computational simulation and validation of tFUS propagation”.

There are multiple missing references. Matzke, Wagner, Lacoutoure … this is sloppy and seriously detracts from the manuscript.

We apologize for this oversight. We have now added all the relevant references.

The EEG pre-processing is insufficient. There is mention of interpolation several times but no details. Data is said to be re-referenced 3 separate times … is this correct? If so, why?

We apologize for the confusing description of EEG pre-processing. We have significantly revised the manuscript in several places (*page 8*) to clarify that data were recorded with a left mastoid reference, were only average referenced once (*page 19*), and electrodes were interpolated only once (*page 19*). Specifically, we restructured and added text to the section *“EEG pre-processing”* to clarify the order in which all processing steps took place. We also explain that data were first down-sampled and high-pass filtered, followed by all EEG segment and electrode rejection, and that artifact subspace reconstruction was then applied to remove non-stationary and noisy segments. Then, in text, we now explain the following:

“After removing the segments identified by ASR, all previously removed electrodes were interpolated, and the data were common average referenced to ensure a zero-sum voltage across channels.”

We hope this clarifies that interpolation was only performed once, after ASR, and right before the only average referencing computed.

Citations are needed for many of the methods for example the use of artifact subspace reconstruction, k-medoids, dynamic time warping, etc. You may know who and why these methods are used but many others will not. There is a line in the methods that this is a "standard approach in inhibitory control electrophysiology" – cite something.There is mention of the approach of Mattia. What is it?

We expanded the explanation of artifact subspace reconstruction (ASR) to include a link to the function used *(clean_rawdata())*, a link to the wiki page describing it, and references to the original publications and usage recommendations/studies as follows:

– The threshold used was 16 standard deviations above the mean projection, not 6.

– We added references and explanation of ASR, including wiki links.

– We included references breaking down the comparison of K-means and K-medoids.

– We added references to a few (of many) papers that have used ICA maps to obtain ERPs for the analysis of response inhibition behavior.

– As we updated our analysis pipeline to a modern convolutional GLM, we have since replaced the matching approach of Mattia. This is explained in the section

“Deconvolution GLM for regression ERPs (rERPs) and separating Stop from Go response ERPs”.

– The data in figure 3 does not appear to be the data of the main logistic regression analysis but is the crux of the paper. The ANOVA before the figure refers to Figure 3 but so also does the ANOVA after Figure 3. Which is it?

In Figure 3. we present the raw data as *p(respond|signal)*, as this keeps in line with previous studies of stop-signal tasks, which we believe makes it more interpretable for the reader. Furthermore, and as we explain with respect to comment #7 below, we have updated the analysis to be a mixed-design ANOVA over the 75% and 95% SSD conditions. Because this approach only uses 2 points per subject, a logistic is unnecessary and we opted for the ANOVA approach over *p(respond|signal)*.

– The data in Figure 3A is from the ANOVA from the logistic regression. I don't think so as these are probabilities and not log odds but are referred to as such. Please confirm. Also, there is another ANOVA done separately for the different SSDs? It is also not clear what data, and how many trials went into the analysis (especially the regression). There is mention of estimating data points. Why was this done? What happened to the actual data?

We note that we have since changed the analysis to use a mixed-design ANOVA over the probabilities themselves (Figure 3A) and limit these analyses to the 75% and 95% SSD conditions. Nonetheless, we take this opportunity to provide clarification on the previous presentation. In the previous analysis, the response inhibition probabilities (per Stop signal delay level) were converted to the negative logit form before fitting, as previously described in the manuscript section *“Statistical analysis”* and in the sub-section *“Behavioral variables”*. When we presented the original *p(respond|signal)* plot, these were the raw probabilities of failing to stop, and we believe this was confused with the log-odds output of the model: the old mixed-effects logistic would have provided log-odds. However, those could also have been converted to probabilities using the inverse logit 1./(1+exp(-Xbeta)), which is how we presented the previous results. We hope this clarification and the new results will provide the reader with a clearer narrative.

– A table outlining the data; means etc. plus the results of all tests would be very helpful in sorting through all the data.

We are unsure which means/test the reviewer is referring to, as test results are described in the updated Results section (e.g., t-values, degrees of freedom, etc). In addition, the SSRTs and *p(respond|signal)* are all shown in Figure 3. Nonetheless, we have included tables to include the results of all tFUS simulations (Table 1) as well as the parameters from the N200/P300 onset latency correlation with SSRT (Table 2). We hope these new tables address the reviewer’s request.

– Finally, the interpretation of the ERP data in Figure 4 looks to be all descriptive. What tests were done to determine latency or peak differences? How were these quantified if at all? Be sure to cite for the use of permutation statistics and were these done only in sensor space but not temporally across the waveforms? How was latency/onset determined and statistically tested?

Given a desire to bring our analysis in-line with standard processing/analysis used for EEG inhibition studies (and per the request of Reviewer 2), we have updated all the EEG (and behavioral) analysis. As we needed to include group factors in the EEG, we opted to drop the permutation statistics and replace them with mixed-design ANOVAs that analyzed the mean ERP amplitudes (P100, N200, P300) in the medial frontal ICA complex. All these tests are described in *“ERP statistics”* as is the convolutional approach to obtaining the ERPs in the methods section. Therefore, the ERP amplitudes were quantified using mixed-design ANOVAs. We note as well in the methods in *“EEG pre-processing”* that these ERPs were obtained from the per-subject (clustered) independent components. In addition, we explain how the N200/P300 onset latency was obtained in section *“Deriving P300 onset latency and regression with change in SSRT”*. We would like to point out that our goals of comparing the SSRT change and correlating with ERPs did not require using a test (e.g., receiver-operator) to obtain the SSRT onset difference between conditions; our aim was to find the time-warped distance between the No-tFUS and Stop-tFUS conditions to correlate the temporal differences with SSRT. We also included a reference showing dynamic time warping is a satisfactory means to obtain an onset latency difference between ERP signals (Zoumpoulaki et al., 2015).

Reviewer #2 (Recommendations for the authors):

We appreciate the reviewer’s enthusiasm and very helpful comments, i.e., “The EEG analyses are advanced, exploiting robust data-cleaning and selection approaches to allow strong inferences for analyses in sensor space”, “Through careful trial-matching and dynamic time-warping, the effects of primary interest – responses evoked by stopping behaviour – could be isolated from those evoked by the go-cue and go-response.” and “.. as the SSRT is inferred from a model fit on the probability of go-responses as a function of the stop-signal delay (more often failing to inhibit go-responses when the stop-signal arrives late), the empirical observation of a latency shift in the closely related P300 ERP is valuable.” The reviewer further noted that “.. by advancing transcranial ultrasonic stimulation to study prefrontal control, this work signifies a paradigm shift towards using interventional tools in cognitive neuroscience.” However, the reviewer also noted that adding comparisons of between group-effects of tFUS was necessary. The reviewer further noted that several methodological details needed clarification and that we should have used more appropriate analyses. We appreciate these recommendations and constructive criticisms, and we have addressed all the concerns raised by the reviewer. We anticipate this has both made the manuscript more readable and strengthened the conclusions of our work. Below we provide a point-by-point reply to each point.

I think adding the direct comparison between rIFG-tFUS and S1-tFUS and between rIFG-tFUS and sham-tFUS to figures 3 and 4 would strengthen the study.

We agree that these analyses and figures would be useful for the reader. To achieve this, we completely restructured our pre-processing and statistical approaches in line with standard approaches to inhibition studies. All these changes are now explained in the *Methods* section, but we briefly recapitulate them here:

– We changed all the cluster permutation tests to mixed-design ANOVAs.

– Because our analyses were focused on a priori hypothesized components, we limited these group analyses to the P100, N200 and P300 ERP components, as well as the N200/P300 latency onset and SSRT correlations. This has substantially strengthened the results, and we appreciate the Reviewer’s suggestion.

– The updated behavioral results are shown in Figure 3, the updated EEG results are shown in Figure 4, and the EEG x SSRT correlations are shown in Figure 5.

It would be wonderful to learn more about the sham condition. From the methods I couldn't work out how this was performed, how the transducer was placed, whether stimulation was applied or only a sound, or how exactly this condition controlled for possible auditory effects.

We agree that more clarification of the Sham setup and logic underlying our control groups was needed. We used the sham condition to control for auditory effects, as we found that coupling the transducer to the scalp while placing it perpendicular to the scalp to prevent sonification still induced an auditory ringing like active tFUS. Our reasoning was that if an auditory effect drove our behavioral results, then we would find similar *p(respond|signal)* effects in the Sham, S1 control and rIFG group. The S1 control group was used as an active sonification control to discern the specificity of the rIFG tFUS. Thus, while it remains uncertain in the literature [1] how to best define a sham control for auditory effects, we still find support for the specificity of our results in that the impact of tFUS was limited to the rIFG group and was not found in the S1 group – which would have had similar auditory sensations as the rIFG group. We added the text below to the Methods (section “*tFUS targeting, setup and parameters”*, pages 10-11):

“In the sham condition, the transducer was placed perpendicular to the scalp where electrode F8 would have been; this is the same electrode that was not used during the active TFUS rIFG group. The basis of this sham was that we (a) wanted to emulate the sound that is cranially detectable during active TFUS, and (b) did not want substantial ultrasound energy transmitted, hence the transducer pointing away and perpendicular to the scalp. The choice of this sham was codified by all the experimenters as producing a detectable sound.”

[1] Braun V, Blackmore J, Cleveland RO, Butler CR. Transcranial ultrasound stimulation in humans is associated with an auditory confound that can be effectively masked. Brain Stimul. 2020 Nov-Dec;13(6):1527-1534. doi: 10.1016/j.brs.2020.08.014. Epub 2020 Sep 4. PMID: 32891872; PMCID: PMC7710976.

A figure with the exact trial timing would be very helpful to understand the timing of the go cue, SSD, exact timing of the tFUS relative to the go and/or stop signal, temporal overlap between tFUS and SSRT and ERP, etc.

We have added a panel to Figure 1 to describe the timing of the “go” and “stop” cues, the “stop” cue onset with respect to SSD and reaction time, the timing of tFUS, and the SSRTs and ERP windows.

Was the tFUS at stop delivered 100 ms before the stop signal (as in Legon 2014, t=-100ms), or at the time of the stop signal (t=0ms)?

tFUS delivery occurred at *t* = 0 ms (see *Behavioral Task and Transcranial Focused Ultrasound design section*, page 7). To clarify this important methodological detail, we have added the following description:

“Simultaneous here means that tFUS was delivered at time = 0 ms, i.e., time synced to either the Go signal or the Stop signal.”

In addition, we hope that the revisions to Figure 1 about trial timing and tFUS timing further clarifies this point (see also reply to comment #12).

Did the stop signal (red square) appear at random times (as suggested in the caption of figure 1), or at one of four discrete SSDs (as suggested by the methods section)?

The SSDs were presented at one of four discrete SSDs. We have edited the Figure 1 caption to clarify that only the timing between fixation and Go was randomized, and that “a red square appeared at one of 4 latencies (SSD) after the Go signal”.

The description of the tFUS methods would benefit from 1) a clearer specification of the intensity together with the other stimulation parameters, and 2) an estimation of indices relevant for safety, such as the MI and estimated thermal rise in the skull and brain or TIC.

We have expanded the description of the tFUS methods and included several of these pieces of information. In addition, we have also included new simulations to present these indices for the undoped silicone we used (in response to Reviewer 1). Specifically, in section “Computational simulation and validation of tFUS propagation”, we present all the intensity parameters (Pmax, ISPPA, ISTPA) and MI for all the simulations in Table 1. In addition, we note these intensities are all in the range of non-thermal modulation.

You report that tFUS improved stopping behaviour. This was only the case for the SSD at 95% of baseline RT, right? It seems that with SSD at 65% of baseRT the performance was impaired (Figure 3A). Although in the text on page 21, the opposing effects at 65% and 95% are together described as '[…] inhibitory performance appear greater at longer SSDs'.

With reference to the old figure 3 (which we have now updated based on new statistical analysis), we believe the reviewer was referring to the median being lower in the 65% for No-tFUS compared to stop-tFUS condition. This was the result of the plotting code incorrectly denoting certain points as outliers automatically. We now have updated the plots, as the included SSDs and analysis has changed, and checked the plotted points as error-bars to ensure correctness. Therefore, in the updated (and previous) analysis, there is a performance improvement at 75% SSD (which we erroneously referred to in the previous version of the manuscript as “65%”), as well as 95%. We thank you for pointing out the confusion and the opportunity to correct it.

On page 17, I didn't quite understand how the ERPs corresponded to the 85% and 105% of mean go RT, while the SSDs corresponded to the 65% and 95% of the baseline go RT. Is this because there was a difference between baseline go RT and task go RT?

We apologize for this confusion: this was a typo carried over several manuscript edits. We have corrected the paper (in section *“EEG Statistics”* and subsections *“Deconvolution GLM for regression ERPs (rERPs) and separating inhibitory Stop from Go response ERPs”*) to clarify that the SSDs are denoted in % of baseline mean Go reaction time, and these SSD values include 15%, 35%, 75%, and 95%.

The EEG analyses might benefit from quantified statistical analyses. Cluster-based permutation tests are described in the methods, and three contrasts are mentioned, but it seems none are reported in the Results section.

In this updated version of the manuscript, we have opted to update nearly all the analyses. These updates include replacing the cluster-permutation tests with field-standard approaches to examining EEG during response inhibition. Specifically, we now examine mean amplitude of ERPs derived from medial-frontal central ICA clusters. We apologize for confusion and sparsity of description in the previous version of the manuscript. As can be seen throughout the methods and EEG Results section, we explain that we quantify the ERP amplitudes using (1) an extraction of peak centered amplitudes around candidate ERPs, and (2) using mixed-design ANOVAs to integrate the reviewer’s proposal to include comparisons across and within groups.

In general, the methods section, especially the description of the statistical inference, could benefit from more clarity and more rigorous and consistent descriptions of conditions and parameters.

We have now added several pages of text in the methods describing how we (1) quantify behavior, (2) justification for certain SSD levels in analysis, (3) description of the convolutional GLM model for ERPs and their utility in dissociating Go and Stop related ERPs, (4) a description of the ERP amplitude extraction, and (5) their analysis using mixed-design ANOVAS that include all groups per ERP component considered. These expansions have been added under sections “EEG-preprocessing”, “Deconvolution GLM for regression ERPs (rERPs) and separating Stop from Go response ERPs”, and “ERP statistics”. We hope the expansion and addition of these sections provides a clear description.

I would recommend using the same y-axis range across the panels of figure 4B. This will allow a direct comparison of the ERPs between no-tFUS and stop-tFUS conditions.

This figure has been updated in line with the new analysis, and the y-axis ranges are now matched.

The discussion might benefit from considering the tFUS timing in relation to the SSRT. How much tFUS was delivered before the SSRT was reached? Especially considering that the effect was only present at SSD = 95%. How does this relate to the known delay between tFUS onset and modulation of spiking activity (e.g. 50-100ms delay)?

First, we would like to reference the reviewer to our response to comment #16 below as it addresses a similar comment. Specifically, we suspect there was confusion in our previous presentation of the *p(responds|signal)* analysis and the original plot had an incorrectly displayed outlier the altered the box plot for the 65% SSD. We also note that this was incorrectly written (we meant to write “75% SSD”), and it has been corrected throughout the manuscript. Second, in the updated analysis, there is also a performance improvement at 75% SSD in the rIFG tFUS group. The new analysis presented in Figure 3A and section *“tFUS to rIFG improves stopping behavior”* is intended to clarify these statistical effects.

With regard to the reviewer’s question of “How does this relate to the known delay between tFUS onset and modulation of spiking activity (e.g. 50-100ms delay)?”, we would appreciate the reviewer pointing us to the paper(s) they are referring to. Upon our own searches, we did not see such a specific delay effect in spiking codified everywhere. In addition, there are also conductance/signal propagation delays between spiking and EEG. This would make it difficult to expand a spiking-related delay to EEG. Finally, we also suspect that the differences in tFUS efficacy between short SSDs and long SSDs could equally result from differentially engaged Go and Stop processes; we discuss this in some length in both the Methods (see *Behavioral Analysis*, *Discussion*, and Bisset et al., 2021).

The discussion mentions that TMS has a resolution in the order of a few millimeters and TUS about 1-2mm. This seems similar, and both are somewhat optimistic. The lateral FWHM of the TUS beam was already 5mm, and the axial/longitudinal FWHM is probably in the order of centimeters.

The discussion of TMS should have read as having a resolution of a few cm, not mm. Furthermore, while several studies have attempted to map the resolution of TMS and tFUS, we suspect that no firm conclusions can be made. Therefore, to prevent an erroneous claim of superior spatial resolution of tFUS over TMS, we removed the argument in the discussion that tFUS necessarily has better resolution than TMS. In addition, we note all the lateral FWHMs for each simulation level in Table 1.

Do you have any ideas on the neurophysiological mechanism through which tFUS led to improved stopping behaviour (or impaired at 65% SSD)?

We would like to comment on these in the reverse order. As we noted in the reply to comment #7 above, performance was improved – not impaired – at 65% SSD (which was actually 75% SSD); performance improved also in the Stop-TFUS signal condition for the 95% SSD condition. We clarify this in the Results section *“tFUS to rIFG improves stopping behavior”* as well as in the reply to comment 16 below. Briefly, there were performance improvements at 75% and 95% SSD for only the rIFG tFUS group. As we also explain in response to comment 21, previous work that came out after our original analysis, has noted that too short of SSDs likely violate assumptions of the independent race model (Bisset et al., 2021) used to analyze *p(respond|signal)* and SSRT, and this may have a large, but unidentifiable impact on inhibition responses. Namely, subjects may not need to cancel movements at short SSDs, but merely ‘not go’. This is why in our updated analysis we only include the 75% and 95% SSD conditions.

In terms of direct neurophysiological impact, we wish to be conservative in making claims about whether the tFUS decreased/enhanced firing of excitatory pyramidal populations in, say, L2/L3 layers versus decreased/enhanced inhibition of inhibitory interneurons in deeper columnar layers. A review of the literature examining cell-type specific responses indicates that evidence for whether tFUS exerts modulation specifically on either cell type or enhances depolarization or suppresses it is mixed. Namely, it remains unclear.

https://www.frontiersin.org/articles/10.3389/fnhum.2021.749162/full

Some conclusions from the discussion do not seem to be strongly supported by evidence. For example, I did not fully understand how you conclude that rIFG tFUS is not involved in P300 amplitude modulation. Also, the conclusion on N200 amplitude relation to response inhibition outcome seems to be based on a visual comparison. Did I miss something? And the same for the conclusion that N100 was modulated in the no-tFUS, but not in other tFUS conditions? Was this also a visual comparison?

We address these here in the order of the reviewers’ questions. Specifically:

– We argue the P300 amplitude is not specifically related to inhibition. We present new evidence for this in the updated manuscript. Based on the reviewer’s request, we included the tFUS x Group comparison of ERP. In doing this, we show a tFUS x Performance outcome interaction of P300 amplitude that does not differ across groups

– In the discussion, we now speculate that this generic interaction across groups suggests (1) an auditory component that (2) relates P300 more to an expectancy outcome violation, rather than an inhibition specific component. Put differently, if the P300 amplitude were inhibition specific, one would predict an interaction that was specific to the rIFG group, given only their behavior was impacted.

– Discussion of the N200 is now updated and is based on the mixed-design ANOVA used to analyze the mean ERP amplitude.

– The N100 is no longer included merely because we changed our analysis approach to the ICA standardized approach for looking at the N200/P300 complex. We would like to emphasize that we now include the P100, and it shows a substantial P100 effect that differentiates successful and failed stopping. However, the P100 did not reveal any Group or tFUS effects.

Together, the analysis of all these ERP components have been updated through the mixed-design ANOVAs. All three ERPs are now presented in the results and considered in the discussion. We hope this updated exposition is much clearer than the previous version.